# The Role of GABA Pathway Components in Pathogenesis of Neurodevelopmental Disorders

**DOI:** 10.3390/ijms26199492

**Published:** 2025-09-28

**Authors:** Ekaterina V. Marilovtseva, Amal Abdurazakov, Artemiy O. Kurishev, Vera A. Mikhailova, Vera E. Golimbet

**Affiliations:** Mental Health Research Center, Kashirskoe Sh., 34, Moscow 115522, Russia; aabdurazakov@edu.hse.ru (A.A.); kurishartt@gmail.com (A.O.K.); voviand.vm@gmail.com (V.A.M.); golimbet@mail.ru (V.E.G.)

**Keywords:** GABA pathway, neurodevelopmental disorders, schizophrenia, epilepsy, ASD, major depressive disorder

## Abstract

γ-aminobutyric acid (GABA), the primary inhibitory neurotransmitter in the central nervous system (CNS), regulates neuronal excitability, synaptic plasticity, and oscillatory activity essential for cognition, emotion, and behavior. Disruptions in GABAergic signaling are increasingly recognized as key contributors to a range of neurodevelopmental disorders (NDDs), including schizophrenia (SZ), autism spectrum disorder (ASD), major depressive disorder (MDD), bipolar disorder (BD), and intellectual disability (ID). In this review, we analyze the data available from the literature concerning the components of the GABA pathway. We describe the main steps of GABA metabolism, including GABA synthesis and release, GABA receptors neurotransmission, GABA reuptake and catabolism, and evaluate their involvement in the pathogenesis of neurodevelopmental disorders. We suggest the possibility of existence of so far undescribed mechanisms which maintain the concentrations of GABA at a relatively physiological level when the function of glutamic acid decarboxylases is compromised by mutations. Searching for these mechanisms could be important for better understanding neurodevelopment and could give a clue for future searches for new therapeutic approaches for treating or alleviating the symptoms of BD and SZ. We also argue that the metabolic stage of the GABA pathway has only a minor direct effect on GABA signaling and rather causes clinical effects due to accumulation of neurotoxic byproducts.

## 1. Introduction

As the primary inhibitory neurotransmitter in the central nervous system, GABA regulates neuronal excitability, synaptic plasticity, and oscillatory activity essential for cognition, emotion, and behavior [1]. Disruptions in GABAergic signaling are increasingly recognized as key contributors to a range of psychiatric disorders, including SZ, ASD, MDD, BD, as well as ID [2,3,4].

The aim of this review is to gather and analyze the data available from the literature, concerning the components of the GABA pathway, and to evaluate their contribution to pathogenesis of various NDDs.

To provide context, we start with the characterization of GABAergic neurons describing their complex classification based on different synaptic profiles underlying cognitive and emotional processes, whose disruption can result in neurodevelopmental conditions. We describe in detail the main steps of GABA metabolism, including (1) GABA synthesis and release, (2) GABA neurotransmission, (3) GABA reuptake, and (4) GABA catabolism. Firstly, we define a role of the main components, crucial for GABA synthesis, with a specific focus on glutamic acid decarboxylases (GADs). We summarize GADs’ functions in the brain and their association with NDDs such as SZ, ASD, epilepsy, etc. Based on the literature analysis, we suggest the possibility of the existence of mechanisms, so far undescribed, realized in early embryogenesis to compensate for heterozygous mutations within the *GAD1* gene. Then, we present evidence implicating GABA receptors in the pathogenesis of MDD, ASD, BD, and SZ through the disbalance in expression rates of different types of GABA_A_Rs and GABA_B_Rs subunits. We further describe in detail a regulation role of the members of the SLC6 family responsible for GABA reuptake and their involvement in the pathogenesis of neurodevelopmental disorders. We especially emphasize the role of GAT1, the key GABA transporter in mammalian CNS, in maintaining the E/I balance and the spatiotemporal specificity of GABA signals in the cortex and other brain regions and provide evidence that GAT1 dysregulation caused by various mutations in the coding gene region SLC6A1 is implicated in a variety of neurodevelopmental disorders. Finally, we describe the components of the final stage of the GABA pathway and consider the therapeutic agents applicable at this stage.

### 1.1. GABAergic Neurons: Classification, Functions, and Implication in Neurodevelopmental Disorders

In embryogenesis, GABAergic neurons (Figure 1) originate from the medial ganglionic eminence (MGE), caudal ganglionic eminence (CGE), and lateral ganglionic eminence (LGE) [5]. Found in both the subcortical structures and prefrontal cortex (PFC), GABAergic neurons constitute up to 25% of all neurons [3]. Depending on the type of synaptic profiles they are associated with, GABAergic neurons may be divided into several subtypes. The first one is basket cells (BCs), associated with axo-somatic inhibitory synapses and providing strong inhibition to the somata and proximal dendrites of glutamatergic pyramidal cells (PyCs) [6], principal granule cells of the dentate gyrus [7], and principal neurons of the basolateral amygdala region [8], thus influencing their output and regulating working memory, cognitive flexibility, emotional responses, and fear processing. BCs are mostly found in cortical layers (L) 3/4 [3], as well as in the hippocampus [9], and the molecular layer of the cerebellar cortex near Purkinje cells [10]. Importantly, BCs are also known to come into contact directly with thalamic afferent fibers and, thus, to be directly engaged in shaping sensory inputs [3]. Based on their electrophysiological properties, BCs can be divided into two subpopulations, fast-spiking (or fast-firing) parvalbumin-positive (PV^+^) BCs and regular-spiking (or regular-firing) cholecystokinin-positive (CCK^+^) BCs [11]. Both PV^+^ BCs and CCK^+^ BCs contribute to the regulation of cortical rhythmogenesis and play a major role in synchronizing and coordinating the functions of the hippocampus [12,13]. According to the other classification, based on their morphological features, BCs are divided into three groups: mostly multipolar large BCs (LBCs), expressing PV, neuropeptide Y (NPY), and calbindin (CB) and are capable of inhibiting the neurons in neighboring columns and in upper and lower layers due to the expansive axonal arborization; vasoactive intestinal peptide (VIP) expressing small BCs (SBCs) with local multipolar (L4), bitufted or bipolar (L2/3) somato-dendritic morphology and frequently branching curvy axons innervating the cells within the same cortical layer; and VIP^–^ nest BCs (NBCs), which share particular morphological features with LBCs and SBCs [14].

The other subtype of GABAergic neurons is PV^+^/CB^+^ chandelier cells (ChCs), associated with axo-axonic synapses, providing inhibition to PyCs’ axon initial segments (AIS), where action potentials are generated. Unlike sparsely branched BCs’ axons, the axons of either multipolar or bitufted ChCs spread horizontally, their terminals arranged in vertical rows of 2 to 15 candlestick-like boutons, with each row innervating the postsynaptic AIS of a single PyC [15]. ChCs are found in the hippocampus, where they constitute up to 4% of all CA1 interneurons [15], PFC, and temporal and visual cortices, with the highest axon terminals’ density observed in L2–6 [16]. Recently, it has been established that in the visual cortex ChCs, receiving inputs from the L5 PyCs, provide the signal to PyCs in L2/3 in response to locomotion and visuomotor mismatch-related arousal [17]. Moreover, action potential discharge in ChCs present in the basolateral amygdala evokes local network activity, which initiates sharp waves (SWs), associated with learning, and memory consolidation and retrieval, by engaging other GABAergic interneurons and PyCs [18].

Axo-dendritic inhibition of PyCs is mostly mediated by somatostatin (SST)-expressing cells, such as double bouquet (DBCs) and Martinotti cells (MCs). The first ones are VIP^+^/calretinin (CR)^+^ regular-spiking interneurons located in L2/3 of the neocortex, characterized by their vertical bundled axons, termed horse-tails, which mostly innervate the shafts of small dendrites and form symmetric synapses with dendritic spines [19], providing disynaptic inhibition among PyCs, thus playing an important role in so-called cortical memory networks [20]. Of note, evidence suggests that DBCs may also target the dendrites of BCs and ChCs, providing a disinhibitory link to the GABAergic system, and selectively target L4. Interestingly, DBCs and PV^+^ GABAergic interneurons are differentially regulated by presynaptic group III metabolic glutamate receptors (mGluR), which increase the frequency of sIPSCs in the former, but suppress it in the latter [21]. With their axons, forming spiny boutons, MCs (whose somata are found in L2–6) reach L1 to target terminal tufts of PyCs, which differs them from DBCs. MCs, capable of inhibiting the dendrites, perisomatic dendrits, and somata of neighboring and even distant columns, are the only cells that mediate cross-columnar inhibition from L2–6 via L1. The other main types of GABAergic interneurons involved in axo-dendritic inhibition are the following: Axon tuft cells (ATCs), confined to L1, their axons following a horizontal trajectory and forming rich arborizations [3]; Bipolar cells (BPCs) with spindle or ovoid somata and vertically oriented dendritic arbors, which are significantly longer in L4–6 BPCs than in L2/3 BPCs; Small button-type neurogliaform cells (NGCs), forming a spherical and highly symmetrical dendritic field and ultra-thin axons with intertwined arborization [22].

The maturation of GABAergic interneurons is one of the key elements initiating critical periods of neuroplasticity—the time-limited windows in brain development during which neural networks are particularly sensitive to sensory and behavioral experiences [23]. Thus, impaired development and maturation of GABAergic interneurons can result in alterations in the cortical and hippocampal circuits associated with severe NDDs, such as SZ and ASD, etc. Indeed, recent studies of clinical cases and animal models have revealed a number of physiological alterations (excitatory/inhibitory (E/I) neurotransmitter imbalance in the hippocampus [24], altered working memory and other functions associated with dorsolateral prefrontal cortex (DLPFC) [25], impaired sleep [26], etc.), appearing in response to a deficit of GABAergic interneuron-dependent inhibition [27], loss [28] or excessive production of GABAergic neurons [29], or compromised function of the main components of the GABA pathway (Figure 2), which are the focus of this paper.

### 1.2. GABA Synthesis and Release

In this section we describe the main components involved in GABA synthesis and release.

#### 1.2.1. Sodium-Coupled Neutral Amino Acid Transporter 1

Mouse models demonstrate that at least in PV^+^ GABAergic neurons the main source of L-glutamate, the immediate precursor of GABA, is astroglia-derived glutamine imported into the cells by Snat1, encoded by *SLC38A1* and detected in proximal dendrites and somata [30]. SNAT1-dependent import of glutamine is a voltage-dependent process driven by Na^+^ (or, alternatively, Li^+^) electrochemical gradient with 1:1 stoichiometry of Na^+^/amino acid cotransport [31]. Playing a pivotal role in regulating GABA replenishment and synaptic vesicular content in GABAergic neurons, SNAT1 is involved in inducing high-frequency oscillations and shaping cortical processing and plasticity [32]. Importantly, upregulation of *SLC38A1* due to the lack of methylation at its promoter was observed in peripheral blood in MDD [33] and in microglia in Rett syndrome [34].

The deteriorating effect of *SLC38A1* overexpression may be explained by a decrease in mitochondria viability and metabolism in response to glutamate overproduction, as well as by neurotoxic effect of elevated NMDAR activation. Logically, the inhibitors of SNAT1-dependent glutamine uptake, such as L-theanine, were proposed as potential therapeutic agents for preventing the onset of ASD [35] as well as for alleviating MDD symptoms [36].

#### 1.2.2. Phosphate-Activated Mitochondrial Glutaminases

Prior to the initiation of GABA synthesis, glutamine is processed to glutamate by a phosphate-activated mitochondrial glutaminase (K-type GLS1 isoform or L-type GLS2 isoform), which generates NH4^+^ as a byproduct. Interestingly, GLSs are positively regulated by Ca^2+^ and inhibited by glutamate in a negative feedback loop [37]. Essential for neurogenesis and neuronal progenitor cell (NPC) survival in embryogenesis [38], the *GLS1* gene is also capable of inducing microglia when overexpressed, thus contributing to neuroinflammation [39]. Examination of postmortem autistic brain samples and those of *GLS1^KO/KO^* mice demonstrated the link between the loss of *GLS1* expression in PFC and ASD symptomatology, including synaptic E/I imbalance and abnormal expression patterns of genes associated with microglial synapse pruning, which were restored by low doses of lipopolysaccharide [40]. At the same time, overexpression of *GLS1* (in particular, its GAC isoform) in the hippocampus, associated with severely impaired spatial and fear memory [41], is observed in MDD models [42], while *GLS1* haploinsufficiency results in hippocampal hypometabolism and schizophrenia resilience phenotype [43], and, thus, has an antipsychotic-like effect [44]. Finally, overexpressed *GLS2* alters mitophagy function in hippocampal neurons, reducing the occurrence of seizure-like events [45].

Altogether, these data indicate that *GLS1*-related pathogenesis of SZ and MDD involve mitochondria dysfunction and neuroinflammation rather than direct dysregulation of the GABA pathway, while in epilepsy and ASD *GLS1* downregulation may indeed affect GABA synthesis and disrupt inhibitory processing in the brain.

#### 1.2.3. Glutamic Acid Decarboxylases

Biosynthesis of GABA from L-glutamate is a one-step reaction catalyzed by paralogous pyridoxal 5′-phosphate (PLP)-dependent glutamic acid decarboxylases GAD65 (encoded by *GAD2*), GAD67 (encoded by *GAD1*) or the isoforms of the latter, such as enzymatically active GAD44, corresponding to the C-terminus of the enzyme [46]. In the human brain, GADs are found in the neocortex, hippocampus, basal ganglia, and cerebellum, with GAD65 being preferentially localized in neuropil granules (axonal boutons or terminals) and GAD67—in neuronal cell bodies and proximal dendrites and nerve terminals [47]. Interestingly, while GAD67 functions to maintain the basal pool of GABA, GAD65 is activated in response to short-term increases in the neurotransmitter demand in synaptic terminals [48]. *GAD3*, the third glutamic acid decarboxylase gene described, is absent in the genomes of higher primates, including humans [49]. Importantly, mouse models demonstrated that *GAD2* complete knockout resulted in higher susceptibility to seizure induction but did not affect GABA concentrations [50], while *GAD1* complete knockout caused neonatal death, severe cleft palate, and respiratory failure, accompanied by dramatic reduction in GABA concentration [51].

Highly homologous (~76% identity), GAD65 and GAD67 differ significantly in the N-terminal region (the first 95 amino acids) [52], which serves as a target for hydrophobic PTMs in case of GAD65, but not GAD67 [53]. Another PTM, known to regulate the activity of both GAD65 and GAD67, is PKA-dependent phosphorylation at Thr91, which activates the former and, vice versa, inhibits the latter [54]. Forming homo- and heterodimers, the two main GAD isoforms are further targeted to synaptic vesicles (SVs), the N-terminus of GAD67 (including Thr131 [55]), and the N-terminal palmitoyl groups of GAD65, playing a crucial role in this process [56]. Once GABA is released to synaptic cleft, GAD65 is clustered at the presynaptic membrane by Neuronal growth regulator 1 (NEGR1) and further removed with newly formed synaptic vesicles [57].

Both *GAD1* and *GAD2* have been found in association with multiple NDDs. Analysis of postmortem MDD specimens showed decreased GAD67/65 levels in the hypothalamic paraventricular nucleus (PVN) [58], which may, at least partly, explain impaired sleep and stress response characteristic to the patients with MDD. Moreover, a particular number of *GAD1/2* SNPs have been identified in patients with NDDs. Recently, a single SZ-associated homozygous polymorphism c.391A>G (p.Thr131Ala) within *GAD1* ORF has been described, disrupting the enzyme’s dimerization and function [55]. A comprehensive analysis of 235 Chinese Han family trios revealed 17 SZ-associated variants of *GAD1*, including ten SNPs in 5′-flanking regions, four SNPs, and one novel in-del in intronic regions, as well as two SNPs in the 3′-UTR. In silico research demonstrated that four of the identified 5′-flanking polymorphisms (rs3762556 (G>C), rs3762555 (G>C), rs6755102 (C>T), rs3791878 (G>T)) can change the affinity of the gene’s promoter for a number of transcription factors [59]. rs3749034 (C>T), another SZ-risk SNP located in the *GAD1* 5′-flanking region [60], was found to affect the methylation status of CpG sites within the gene’s putative promoter, downregulating *GAD1* expression and increasing GAD25/GAD67 ratio in the hippocampus and DLPFC of patients with SZ [61,62]. On a clinical level, rs3749034 (C>T) is associated with negative symptoms and early-onset SZ, as well as with BD and MDD [63]. *GAD2* expression rate was found to be reduced by 40% in the primary auditory cortex of SZ patients, indicating that the gene might contribute to auditory hallucinations occurrence [64], and GAD65 autoantibodies in serum and cerebrospinal fluid were attributed to catatonia in patients with prediagnosed chronic SZ and proposed as the condition’s biomarker [65]. *GAD1/2* are also dysregulated in ASD; for instance, some patients exhibited a ~40% decrease in the level of *GAD1* mRNA in Purkinje cells [66], which is, at least in part, due to the enrichment of 5hmC and MECP2 at its promoter [67]. In this regard, it is worth mentioning that prenatal immune activation due to maternal infections may result in increased levels of 5mC and 5hmC at the *GAD1* promoter and that of 5mC at the *GAD2* promoter in PFC. This, in turn, results in suppression of *GAD1/2* promoters by elevated MECP2 binding and, on a global level, in compromised PFC function, manifesting in cognitive abnormalities, impaired working memory and social interaction, as well as other symptoms typical for ASD and SZ [68].

Finally, there are multiple lines of evidence confirming the role of *GAD1/2* in epilepsy pathogenesis. Several bi-allelic *GAD1* variants associated with early-infantile onset epilepsy and severe neurodevelopmental delay have been identified: homozygous c.87C>G (p.Tyr29*), c.568delC (p.Gln190Serfs ter11), c.971T>G (p.Phe325Cys), and c.1040C>T (p.Thr347Met) and heterozygous c.1591C>T/c.670delC (p.Arg531*/p.Leu224Serfs*5) among them [69]. Autoimmune-associated epilepsy (AAE), a drug-resistant disorder, has also been described in the literature. It manifests in refractory focal seizures and, on the histological level, in infiltration of the temporal lobe by CD8+ cells, and its pathogenesis is tightly linked to the GAD65 autoantibodies [70]. Recently, several polyphenols have been described as potential therapeutic agents in treating GABA deficiency-associated epilepsy. In particular, one of them, pterostilbene (PTE, 3,5-dimethoxy-4′-hydroxystilbene), was shown to upregulate *GAD1* expression and, hence, to increase GABA synthesis in the hippocampal PV^+^ GABAergic neurons, as well as in the PFC in rats [71] and mice [72]. The other natural compound, honokiol, was found to stimulate GAD65 activity in the hippocampus of mice [73]. Also, intracerebral electroacupuncture in combination with pentobarbital significantly enhanced *GAD1* expression in rats and was highly efficient in reducing seizures [74].

Based on the literature analysis and given the importance of GABA for neurogenesis and brain metabolism, we suggest that in early embryogenesis the cell might possess so far undescribed mechanisms to compensate for heterozygous mutations within *GAD1* gene, encoding the key GABA synthesizing enzyme, in order to maintain GABA concentrations at the levels essential for surviving. These mechanisms may include *GAD2* and/or fetal isoforms of *GAD1*, as well as upregulation of *GAD1* expression by epigenetic mechanisms, such as H3K4 trimethylation, which was found to be reduced at *GAD1* regulatory sequences in SZ [75]. This assumption is indirectly confirmed by the fact that, although associated with epilepsy and severe developmental delay, the bi-allelic *GAD1* mutations (even those resulting in truncated GAD67 lacking the catalytic domain), described so far, are not lethal. At the same time, ubiquitous downregulation of *GAD1* due to perturbations in promoter methylation results in complete allele LOF and significantly decreased transcripts; in turn, GABA concentration dramatically drops and so does the efficiency of inhibitory neurotransmission, inevitably leading to impaired neurodevelopment and subsequent disorders. Apparently, introducing large deletions into *GAD1* may have a similar effect, making animal models with *GAD1* haploinsufficiency less informative for studying the clinical effects of GAD67 mutations.

#### 1.2.4. Vesicular GABA Transporter/Vesicular Inhibitory Amino Acid Transporter

Once synthesized, GABA is packaged into synaptic vesicles (SVs) which are then trafficked to the presynaptic membrane to fuse with and to release the neurotransmitter to the synaptic cleft in response to the increase in intracellular Ca^2+^ concentration (Figure 3) [76,77]. The central role in both processes is played by vesicular GABA transporter/vesicular inhibitory amino acid transporter (VGAT/VIAAT), encoded by *SLC32A1*, the only known member of the SLC32 family; interestingly, VGAT-dependent loading of GABA into the SV requires for electrochemical luminal proton gradient [78,79]. The VGAT molecule consists of nine transmembrane domains (TMs), a long N-terminal domain facing the cytoplasm, and the C-terminus facing the lumen [80]. Interestingly, VGAT is also known to transport glycine, another inhibitory neurotransmitter [81], and to co-localize with glutamate transporters VGLUT1 (*SLC17A7*) [82] and VGLUT2 (*SLC17A6*) [83] on SVs. *SLC32A1^−/−^* mice exhibit complete stiffness by day E17.5 and die in utero after day E18, these observations confirming the indispensability of VGAT in development [79]. In humans, mutations in *SLC32A1* have been identified in association with various epileptic disorders. The study of a large cohort of Australian patients with genetic epilepsy with febrile seizures plus (GEFS+) or idiopathic generalized epilepsy (IGE) revealed eight rare inherited heterozygous missense mutations in *SLC32A1* (c.127G>T (p.Gly43Cys), c.788T>C (p.Val263Ala), c.989T>C (p.Met330Thr), c.1333C>T (p.Leu445Phe), c.1382G>A (p.Gly461Asp), c.1391C>G (p.Thr464Arg), c.1393G>A (p.Gly465Ser), c.1403T>C (p.Leu468Pro)), affecting the structure of the transporter’s TMs and N-terminus and apparently associated with reduced vesicular GABA transport [84]. A link between four *SLC32A1* heterozygous de novo mutations (c.271G>A (p.Ala91Thr), c.787G>A (p.Val263Met), c.806T>C (p.Leu269Pro), c.965T>G (p.Phe322Cys)) and epileptic encephalopathy with infantile seizure onset and moderate-to-severe ID has also been described. Importantly, located at the N-terminus, the first mutation increases presynaptic GABA release probability and, hence, the stimuli frequency, while the other three found in the α-helices lining the GABA transport pathway affect the substrate uptake. Depending on the mutation, the patients predominantly showed either focal and focal impaired awareness seizures, or tonic–clonic seizures with myoclonic seizures and status epilepticus with suspicion of myoclonic atonic epilepsy in case of c.787G>A (p.Val263Met) [85]. Similar to GAD65/67, VGAT is implicated in ASD [86] and SZ [87], its expression being decreased in the PFC of adult BTBR mice and in postmortem specimens, respectively.

Based on clinical data, we speculate that, at least in embryogenesis, malfunctioning of VGAT is more deteriorating for the GABA pathway and neurodevelopment than that of *GAD1/2*, and that *SLC32A1* polymorphisms severe enough to cause ASD or SZ might be lethal and that is why they have not yet been identified.

Interestingly, there are a number of other factors, including some signaling pathways, that govern GABA release. One of them is the RAS–ERK signaling cascade, which is directly involved in pathogenesis of various health conditions called “RASopathies”, some of which manifest in ID and learning disabilities [88]. Recently, it has been shown that while in excitatory neurons, hyperactivation of RAS–ERK signaling due to the SNPs (such as p.Gly12Val) within the *KRAS* is cytotoxic and leads to cell death, in GABAergic neurons it increases the risk of spontaneous GABA release [89]. The latter is, apparently, due to phosphorylation of synapsin I with its subsequent dissociation from SVs [90]. At a physiological level, this results in increased sIPSC frequency, as well as impaired long-term potentiation at CA3–CA1 synapses in the hippocampus that can be rescued by picrotoxin treatment [89]. This brings us to the conclusion that when designing new diagnostic protocols and therapeutic agents for treating NDDs associated with dysregulated GABA traffic and release, the whole spectrum of its regulators should be taken into account.

### 1.3. GABA Neurotransmission

In the neural networks of the brain, GABA functions as the primary inhibitory neurotransmitter, acting on both ionotropic (GABA_A_R, or GABRA) and metabotropic (GABA_B_R, or GABRB) receptors, both of which are widely distributed in the cerebral cortex, hippocampus, basal ganglia, thalamus, cerebellum, and brainstem [91].

#### 1.3.1. GABA_A_ Receptors

GABA_A_Rs are heteropentameric ligand-gated Cl^–^ channels, detected in hippocampal and PFC PyCs, cerebellar granule cells, thalamic relay cells [92], and a subset of dopaminergic neurons in substantia nigra [93]; serotonergic and GABAergic neurons are also believed to possess distinct subtypes of GABA_A_Rs [94] and to be involved in the realization of the most prominent inhibitory pathway in the CNS. Despite a significant diversity of GABA_A_R’s subunits expressed in the human organism, mostly in a tissue-specific manner (six α (1-6), three β (1-3), three γ (1-3), three ρ (1-3), and one each of the δ, ε, π, and θ), the main adult isoform of the receptor is built up by 2 α1-, 2 β2-, and 1 γ2-subunits [95]. While the α1β2γ2 GABA_A_R subtype, accounting for ∼43% of all GABA_A_Rs, is expressed by GABAergic interneurons and pyramidal cells more or less ubiquitously within the brain [96], the other GABA_A_R subtypes are specific for particular brain structures: α1β2,3γ2 GABA_A_R is characteristic for aspiny cells and α2/α3β2,3γ2 for PyCs of the amygdala [97], α4βδ GABA_A_R [98] and α5β3γ2/γ3 [99] are found in hippocampal PyCs, and α6-containing GABA_A_Rs are restricted to differentiated cerebellar granule cells [100]. Interestingly, GABA_C_Rs, consisting solely of ρ-subunits, are also classified as one of the subgroups of GABA_A_R’s isoforms [101]. Depending on the class of the subunits it is made of, GABA_A_R provides either phasic inhibition (a short-term hyperpolarization of the postsynaptic membrane occurring after GABA release), or tonic inhibition (a long-term hyperpolarization of the postsynaptic membrane by low ambient levels of GABA) [95]. Also, an abundance of phosphorylation sites has been identified in various GABA_A_R subunits, governing the receptor’s activity [102]. Importantly, GABA_A_R is a target for benzodiazepines, which potentiate the effect of GABA [103] and are known to reduce the symptoms of anxiety and SZ when used in combination with particular AEDs [104]. Resveratrol is another agent known to potentiate GABA_A_R-mediated inhibitory neurotransmission [105,106] and suggested as a promising AED [107].

Imbalanced expression of different GABA_A_R subunits has been detected in BD, SZ, and MDD, e.g., postmortem analysis revealed upregulation of β3 subunit and downregulation of α1 and α2 in the SZ lateral cerebellum [108,109], and life [^11^C]Ro15-4513 position emission tomography (PET) imaging demonstrated significant reduction in the α5 subunit in the SZ hippocampus, but not in patients partaking in antipsychotics [110]. In addition, N-glycosylation of β1 and β2 subunits is altered in the SZ cortex, which may result in disrupted GABA_A_R assembly and function [111]. Furthermore, impaired transmission at α6-GABA_A_Rs correlates with dysregulation of dopaminergic system and is characteristic for SZ patients; furthermore, positive allosteric modulators highly selective for α6-GABA_A_Rs were shown to reduce positive and negative symptoms and cognitive impairment in animal models for SZ [112]. Also, the expression of β3 subunit of GABA_A_R was reduced in peripheral blood mononuclear cells in patients with BD, which may be used as a diagnostic marker [113]. Several GABA_A_R subunit polymorphisms were also reported in GEFS+ (e.g., *GABRA1* c.274T>C (p.Phe92Leu)), MAE (e.g., *GABRB3* c.851T>C (p.Leu284Pro)) [114], Lennox–Gastaut syndrome (e.g., *GABRB3* c.358G>A (p.D120N)) [115], and MDD with the risk of suicide (*GABRA6* 3′-UTP SNP rs3219151 (C>T)) [116], but not in SZ [117]. Pathogenesis and clinical manifestation of a particular number of IDs, including those associated with chromosomal aberrations, are also known to be linked to impaired expression of GABA_A_R subunits. In Prader–Willi syndrome (PWS), a genetic disorder characterized by mild-to-moderate ID, impaired behavior, and depression [118], paternal alleles of *GABRA5*, *GABRB3*, and *GABRG3* are missing due to the partial deletion of an imprinted region on paternal chromosome 15 (15q11–13) [119]. Given that all three GABA_A_R subunits are associated with the hippocampus [99], the loss of a half of their alleles may explain poor short-term memory, another known symptom of PWS [120]. Furthermore, according to the in vivo data obtained by single–voxel proton magnetic resonance spectroscopy (1H-MRS), GABA levels are also reduced in the PWS brain (at least, in the parieto-occipital lobe) [121], hinting at the possibility of existence of a feedback loop regulating either GAD1 expression or GAD67 enzymatic activity in GABAergic neurons in response to the fluctuations of GABA_A_Rs’ content in PyCs. Impaired GABA pathway is also typical for Fragile X syndrome (FXS), an ID caused by an expansion of the CGG repeat within the 5′-UTR of Fragile X messenger ribonucleoprotein 1 coding gene *FMR1* [122]. FXS mice models demonstrated 35–50% decrease in expression rates of *GABRA1*, *GABRA3*, *GABRA4*, *GABRB1*, *GABRB2*, *GABRG1*, and *GABRG2* in the cortex [123], as well as a four-fold decrease in Gabrd subunit in the hippocampus [124]. Interestingly, [^11^C]flumazenil PET performed on FXS patients identified a significant reduction in the non-displaceable binding potential (BP_ND_) of GABA_A_Rs (reflecting their availability), throughout the brain, with the highest level of BP_ND_ reduction—up to 17%—detected in the thalamus [125]. Dysregulation of several other components of the GABA pathway have also been found in association with FXS, GAD67/65, GABA transporters GAT1 and GAT4, and GABA catabolizing enzymes among them [122]. Of note, in contrast to PWS and FXS, in Down syndrome (DS), GABA inhibition (mainly tonic) is enhanced. This phenomenon may be explained by the presence of an extra chromosome 21 and, hence, additional copies of *OLIG1*, *OLIG2*, and *DYRK1A* [126], well-known regulators of GABAergic interneurons production [127] and synaptic plasticity pathways [128]. Given the abovementioned data, it was suggested that treatment with basmisanil, a α5-GABA_A_R-specific negative allosteric modulator, could have a positive effect on cognition in patients with DS, but unfortunately, the clinical trials failed [129]. Nevertheless, therapeutic strategies targeting other components of the GABA pathway or the proteins modulating GABA neurotransmission may be effective in alleviating DS symptoms. One of the promising targets for such therapies is G-protein-activated inwardly rectifying potassium channel 2, GIRK2, whose gene is located on chromosome 21 and whose function is linked to that of GABA_B_ receptors [130,131].

#### 1.3.2. GABA_B_ Receptors

GABA_B_R is a heterodimeric G protein coupled class C receptor, expressed in glial cells [132] and somato-dendritic compartments of neocortical and hippocampal PyCs [133], serotonergic neurons in dorsal raphe nucleus (DRN) [134], dopaminergic neurons in substantia nigra [135], and other types of neurons. Each of the two receptor’s subunits, GABAB1 and GABAB2, contains an extracellular Venus Flytrap (VFT) domain, a heptahelical TM, and an intracellular C-terminal region which contributes to heterodimerization and serves as a binding site for several regulatory proteins. When activated by GABA or baclofen, GABA_B_R undergoes conformational changes which result in G protein dissociation into Gα_i/o_ and Gβγ subunits, the former inhibiting adenylyl cyclase (AC) and cAMP signaling, thus preventing vesicle fusion and neurotransmitter release; meanwhile, the latter directly binds to VGCC, inhibiting Ca^2+^ influx, and to GIRK(1-4) channels, triggering K^+^ efflux and contributing to the postsynaptic membrane hyperpolarization. Furthermore, in cellulo GABA–GABA_B_R pathway was found to stimulate phosphorylation of ERK1/2 protein kinases, which subsequently activate a number of transcription factors, involved in neurogenesis and memory formation [136]. Hence, GABA_B_Rs play an essential role not only in regulating E/I balance in the adult brain, but also in coordinating the early stages of neurogenesis, and its dysregulation may have dramatic consequences for cognition and behavior. Indeed, genetic and postmortem studies have confirmed the association between GABA_B_Rs deficits and NDDs such as ASD, BD, SZ, and MDD. In lateral cerebella, for example, the levels of GABBR1 (which is located in SZ-associated loci) and GABBR2 are significantly reduced in cases of BD, SZ, and MDD [137], at least partly due to their attenuation by SZ-related microRNAs [138]. Additionally, exome sequencing identified the rs10985765 SNP (p.Thr611Ala/Pro, p.Thr771Ala/Pro, or Thr869Ala/Pro, depending on the isoform) within *GABBR2* as a potential risk factor for treatment-resistant SZ [139]. Finally, the actin-binding protein SHROOM4 (SHRM4) (encoded by *KIAA1202*) has recently been shown to play a role in regulating the receptor’s activity by providing its synaptic localization [140]. This finding makes GABA_B_Rs a key player in pathogenesis of Stocco Dos Santos syndrome [141] and the NDD caused by Xp11.2 microduplication [142], both of which are associated with disrupted *KIAA1202* and characterized by mild or mild-to-severe ID, seizures, anxiety, and impaired behavior.

Similar to GAD67/65, GABA_A_Rs and GABA_B_Rs subunit polymorphisms tend to correlate with epilepsy. Meanwhile, MDD, ASD, BD, and SZ seem to be associated with the disbalance in expression rates of different types of GABA_A_Rs and GABA_B_Rs subunits. Such disbalance may prevent different subtypes of the receptors from maturation, assembly, and incorporation into the synapses, and subsequently result in decreased inhibitory neurotransmission in one cell type of neurons and increased inhibitory neurotransmission in the others. This, in turn, may lead either to impaired neurogenesis, or to altered E/I balance in different structures of the adult brain. In the former case, dysregulated expression of the GABA receptor subunits may be either due to the presence of SNPs in their regulatory elements, altering their methylation status or affinity to transcription factors, or due to the lack of particular transcription factors; in the latter case, epigenetic regulation must play a central role.

### 1.4. GABA Reuptake

Once released, GABA does not undergo enzymatic breakdown within synaptic cleft, but is eliminated through reuptake via specialized transporters, belonging to the solute carrier 6 (SLC6) family of neurotransmitters:sodium symporters. Only four members of the SLC6 family, SLC6A1/GAT1, SLC6A11/GAT3, SLC6A13/GAT2, and SLC6A12/BGT1 [143], are known to effectively transport GABA by its symport with 2 Na^+^ ions and one Cl^–^ ion. Each of these four proteins consists of 12 helical TMs, connected by intra- and extracellular domains, with TMs 1, 3, 6, and 8 forming the orthosteric substrate binding pocket [144], which is a target for the majority of GAT-specific inhibitors, such as (S)-SNAP-5114 ((S)-1-[2-[Tris(4-methoxyphenyl)methoxy]ethyl]-3-piperidinecarboxylic acid) [145] and a highly selective and potent isatin derivative 1H indole 2,3-dione [146]; of note, the substrate binding pocket is capped from the extracellular side by two amino acid residues from TM3 and TM6, respectively (Tyr140 and Phe294 in case of GAT1; Tyr129 and Phe288 in case of GAT2; Tyr147 and Phe308 in case of GAT3), their mutations leading to dramatic reduction in GABA transport efficiency [147,148]. To note, similar to other SLC6 proteins, GABA transporters tend to oligomerize [149].

Below, we describe in detail the regulation role of the members of the SLC6 family and their involvement in the pathogenesis of neurodevelopmental disorders.

#### 1.4.1. Sodium- and Chloride-Dependent GABA Transporter Type 1

GAT1 is the key GABA transporter in mammalian CNS. Though mostly localized in astrocytes and presynaptic terminals of GABAergic neurons (Figure 4) [150], it is also found in striatal dopaminergic (DA) neurons [151]: taken up to DA neurons by GAT1, GABA is loaded into SVs by the monoamine transporter VMAT2 (the product of *SLC18A2* gene) [152] and is further co-released with dopamine in order to tune striatal output [153] and prevent its hyperactivation, observed in SZ patients [154].

Notably, up to 50% of membrane-associated GAT1 molecules are immobile, anchored to the actin cytoskeleton due to the ability of GAT1 C-terminus to interact with scaffolding proteins [155], with disruption of these interactions increasing the lateral mobility and functional efficiency of GAT1. Given its participation in regulating both phasic and tonic inhibition [156], GAT1 serves as a multifunctional and dynamically regulated transporter, critical for maintaining the E/I balance and the spatiotemporal specificity of GABA signals in the cortex and other brain regions. Therefore, *SLC6A1* dysregulation is implicated in a variety of neurodevelopmental disorders. In a large-scale functional screen of 213 unique *SLC6A1* SNPs, including 67 de novo mutations, approximately 59% exhibited severe LOF (defined as <82.1% GABA uptake compared to wild-type) and 11.7% moderate LOF. Notably, 98.3% of individuals with the de novo variants were diagnosed with developmental delay (e.g., c.863C>T (p.Ala288Met), c.1070C>T (p.Ala357Val), c.1342A>T (p.Lys448*), etc.), 83.6% with epilepsy (predominantly, epilepsy with myoclonic-atonic seizures, EMAS) (e.g., c.881_883del (p.Phe294del*5), c.815_817delTCA (p.Ile272del*4), c.491G>A (p.Cys164Tyr), etc.), and 54.5% with ASD (e.g., c.896G>T (p.Gly299Val), c.1017C>G (p.Phe339Leu), c.1396_1379 delGG (p.Gly457Hisfs*10), etc.) [157,158]. ClinPred and other missense severity scores confirmed high sensitivity of GAT1 to amino acid substitutions (particularly within TMs 1–10, known to build the substrate binding pocket and maintain core structural architecture), which explains *SLC6A1* sevenfold enrichment for pathogenic missense variants over truncating mutations in neurodevelopmental cohorts, despite the relatively small coding size of the gene (599 amino acids). These findings support a unifying haploinsufficiency model, wherein a single functional allele fails to maintain adequate GABAergic signaling, resulting in a broad spectrum of NDDs. Furthermore, surface expression assays revealed that the majority of LOF missense variants result from misfolding, ER retention, and defective membrane trafficking [159], while the remaining ones exhibit impaired transporter function despite GAT1 correct localization [157]. Importantly, three *SLC6A1* de novo variants—c.277G>A (p.Ala93Thr), c.631C>T (p.Arg211Cys), and c.1484G>T (p.Trp495Leu)—were identified in association with both epilepsy and SZ [157,160]. The mechanism underlying SLC6A1-related SZ is still to be unveiled, although most likely it implicates the protein misfolding and subsequent degradation, as at least in DLPFC of SZ patients the levels of GAT1 were decreased [161]; it might also be speculated that SLC6A1-related SZ pathogenesis involves impaired GABA uptake by DA neurons.

The best-known inhibitor of GAT1 is an antiepileptic drug (AED) tiagabine (Gabitril™), approved by the FDA almost three decades ago [162]. Recently, low concentrations of betaine (which is transported through the blood–brain barrier by BGT1/SLC6A12) have also been shown to inhibit GAT1, the transporter’s activity being restored by increasing the concentrations of extracellular GABA or betaine itself [163]. Finally, recent studies have also demonstrated a moderate effect of 4-phenylbutyrate in reducing seizures (but not in mitigating cognitive impairment) in patients with *SLC6A1* and *SLC6A11* haploinsufficiency, caused by a microdeletion on 3p25.3 [164] and associated with ID, epilepsy, and multiple congenital abnormalities [165].

#### 1.4.2. Sodium- and Chloride-Dependent GABA Transporter Type 2

GAT2, encoded by *SLC6A13*, exhibits low expression levels in the brain and is predominantly localized to leptomeningeal cells (pia and arachnoid mater), the ependymal lining of the ventricles, and the endothelium of select blood vessels (Figure 4); so, under physiological conditions, GAT2 does not participate in GABA reuptake from the synaptic cleft and has no influence on synaptic transmission, rather facilitating the movement of taurine from the brain into the bloodstream across the blood–brain barrier and participating in regulating GABA concentrations or osmotic pressure within the cerebrospinal fluid (CSF) [166]. Nevertheless, experiments in mice demonstrated possible association between depression-like states and decreased *SLC6A13* expression [167].

#### 1.4.3. Sodium- and Chloride-Dependent GABA Transporter Type 3

In contrast to rat cortexes, where GAT3 was found to be expressed exclusively in glial cells [168], in human cortexes it was detected both in astrocytes and perikaryal and dendrites of neurons (mostly, PyCs) (Figure 4) [169]. Importantly, later studies demonstrated that GAT1 and GAT3 co-express in thalamic astrocytes, where GAT1 is located closer to the synaptic cleft and is involved in GABA clearance during phasic transmission, whereas GAT3 is positioned more distally and participates in regulating tonic inhibition [170]; this spatial segregation of the two GABA transporters implicates the involvement of GAT3 in preventing GABA from diffusion at long distances, which makes it particularly relevant during prolonged or low-frequency neuronal activity when maintaining reduced perisynaptic GABA levels over extended periods of time is essential [171]. Impaired SLC6A11 function is mostly associated with epilepsy. For example, the polymorphism rs2272400 (c.1572C>T), potentially influencing the mRNA stability and translation kinetics, is associated with temporal lobe epilepsy in patients with a history of febrile seizures and is linked to resistance to AED therapy [172]. Moreover, GAT3 is also implicated in the pathogenesis of mood disorders: experiments in Sprague-Dawley rat models with congenital helpless behavior (cH) showed significant SLC6A11 downregulation [173]. Of note, another study performed in rats demonstrated the role of the transporter in maintaining astrocytic plasticity in the central nucleus of the amygdala and in responding to withdrawal from ethanol exposure, the latter manifesting in upregulation of GAT3 and changes in GAT3-sensitive tonic GABA currents [174]. Altogether these data indicate that it is dysregulation of tonic inhibition that underlies pathogenesis of GAT3-related disorders. Upregulation of GAT3 was also observed in SZ DLPFC and considered to be the result of a compensative mechanism activated in response to GAT1 deficit [161].

#### 1.4.4. Sodium- and Chloride-Dependent Betaine/GABA Transporter

Despite its low expression rate in brain and extrasynaptic localization, BGT1 (SLC6A12) betaine-transporter is also involved in GABA reuptake in CNS and plays a role in pathogenesis of some NDDs [175]. For example, a study of a relatively large European cohort of patients revealed the association between *SLC6A12* haplotype rs542736(G)–rs557881(A) (apparently, resulting in its overexpression) and mesial temporal lobe epilepsy with hippocampal sclerosis [176], which makes BGT1 a potential target for anti-seizure therapy [177]. Additionally, *SLC6A12* SNP rs216250 was shown to correlate with the Scale for the Assessment of Negative Symptoms (SANS) scores in a Korean population, which allows us to presume that BGT1 might be genetically associated with SZ [178]. Hypothetically, the involvement of *SLC6A12* in the pathogenesis of NDDs may be indirect and explained by the role of betaine in modulating GAT1 activity [163].

### 1.5. GABA Catabolism

#### GABA Transaminase

In the final stage of its pathway, GABA undergoes deamination and subsequent oxidation to form succinic acid. The first reaction, resulting in succinic acid semialdehyde (SSA), is catalyzed by GABA transaminase (GABA-T or GABA-AT, encoded by *ABAT* gene) that belongs to class-III of PLP-dependent aminotransferase family and exists in the form of a homodimer [179]. GABA-T acts to transfer the amino group from GABA to α-ketoglutarate, thus replenishing L-glutamate, which in neurons may further be used as a substrate for GAD1/2 in a new round of GABA synthesis [180]. Interestingly, located in the mitochondrial matrix, GABA-T also promotes dNDP conversion into dNTP, hence, providing the organelle with the material for mtDNA synthesis [181]. Of note, GABA-T is a well-known therapeutic target for 4-aminohex-5-enoate, γ-vinyl-GAB, or vigabatrin, an AED, whose action is based on irreversible inhibition of the enzyme [182]. Astrocytic GABA-T is vital for cognitive function, its suppression in the dentate gyrus leading to increased tonic inhibition, reduced neuronal excitability, and impaired long-term memory [183]. Inherited GABA-T deficiency is an ultra-rare autosomal recessive neurometabolic disorder, which causes infantile death in at least a half of the cases; the main symptoms of the disorder are hypotonia, epileptic encephalopathy, choreoathetosis, hyperreflexia, hypersomnolence [184], and dysmyelination of sub-cortical white matter [185], as well as depleted mitochondrial population [181]. With only a few cases of GABA-T deficiency described in the literature, the information available on the *ABAT* mutations associated with the disease is scarce. The short list of the identified pathogenic gene variants is the following: homozygous c.316G>A (p.Gly106Ser) [186], c.1490G>A (p.Arg497His) [187], c.888G>T (p.Gln296His) [188], c.631C>T (p.Leu211Phe), and c.1129C>T (p.Arg377Trp) and heterozygous c.454C>T (p.Pro152Ser)/c.1393G>C (p.Gly465Arg), c.862-2A>G/c.1505T>C (p.Leu502Pro), c.275G>A (p.Arg92Gln)/c.199-?_316+?del (p.Asn67ValfsTer8), and c.659G>A (p.Arg220Lys)/c.1433T>C (p.Leu478Pro) [184]. No treatment is available for this condition.

### 1.6. Succinate-Semialdehyde Dehydrogenase

The last step of GABA metabolism is SSA conversion into succinic acid by NAD^+^-dependent succinate-semialdehyde dehydrogenase (SSADH, encoded by *ALDH5A1*). The localization of the enzyme within the mitochondria allows the product of the reaction to immediately enter the Krebs cycle [189], which provides the cells with α-ketoglutarate, the precursor of glutamate and glutamine. A single monomeric SSADH consists of three domains—an N-terminal NAD^+^-binding domain (amino acid residues 48–173, 196–307, 509–524), a catalytic domain (residues 308–508), and an oligomerization domain (residues 174–195 and 525–535) built up by three β-sheets [190] and responsible for the formation of an active homotetrameric enzyme [191]. Under normal conditions in vivo, about 0.16% of GABA is used by NADP^+^-dependent SSA reductase (SSR) as a substrate for γ-hydroxybutyrate (GHB), which acts to inhibit dopamine release, indirectly increase serotonin turnover in the striatum and mesolimbic areas, and depress AMPA/kainite ePSP [192]. Therefore, elevated concentrations of GHB, detected in SSADH deficient neurons, are highly cytotoxic and are associated with a rare (1:460,000) autosomal recessive condition SSADHD (SSADH deficiency) [193]. SSADHD manifests in aciduria as well as varying degrees of intellectual disability, ASD, anxiety, speech deficits and, in some cases, seizures and/or psychosis in combination with auditory and visual hallucinations with an onset in adolescence or early adulthood [194]. SSADHD-associated SNPs described to date are relatively benign c.269T>C (p.Val90Ala) and c.538T>C (p.His180Tyr), exacerbated when the other allele contains a pathogenic mutation, e.g., c.763A>G (p.Asn255Asp) or c.278C>T (p.Cys93Phe), affecting NAD^+^ binding [195], heterozygous combination c.1226G>A (p.G409D)/c.1498G>C (p.V500L) resulting in the loss of enzymatic activity [196], protein destabilizing homozygous c.1321G>A (p.G441R) [197], and a double heterozygous combination c.526G>A (p.G176R);c.538C>T (p.H180Y)/c.709G>T (p.A237S);c.1267A>T (p.T423S), which both abolish SSADH enzymatic activity and destabilize its tertiary structure increasing degradation of the enzyme [198]. One of the proven effective anti-SSADHD therapeutic approaches implies long-term treatment of the patients with vigabatrin, which results in remarkable reduction in GHB concentrations in CSF and urine and significant improvement of communicative skills [199]. Among the most promising alternative anti-SSADHD therapeutic agents are NCS-382 [200] and PI3K/mTOR inhibitor XL-765 [201], which demonstrated high efficacy in vitro and in vivo in mouse models.

## 2. Conclusions and Future Perspectives

The GABA pathway (Figure 5) can be roughly divided into two main stages—the metabolic stage, also known as “GABA shunt” [202], involving glutamate and GABA synthesis, as well as GABA catabolism, and the signaling stage, involving GABA traffic, release, interaction with the receptors, and reuptake by GATs. Both dysregulation of the signaling stage and impairment of GAD67/65 function have a direct impact on the GABA pathway. Interestingly, involvement of the GABA receptors GABA_A_Rs and GABA_B_Rs and the key GABA synthesizing enzyme GAD67 in SZ and, apparently, ASD and MDD pathogenesis is unlikely due to the mutations in their ORFs, but rather due to dysregulation of their expression caused either by functional mutations in gene regulatory elements, or by epigenetic perturbations and/or disrupted function of particular transcription factors. In case of GABA_A_Rs and GABA_B_Rs, this may result in imbalanced production of different subtypes of the GABA receptors and impaired neurogenesis or altered E/I balance within the brain. As for GAD67, we find it plausible that in early embryogenesis additional mechanisms exist, compensating even for polymorphisms in the ORF of its gene, *GAD1*, and allowing for maintaining the concentrations of GABA at a relatively physiological level. These mechanisms may potentially involve GAD65, fetal *GAD1* isoforms, or upregulation of *GAD1* expression (which may be prevented by the presence of SNPs in the regulatory elements). Searching for these mechanisms could be important for better understanding neurodevelopment and could give a clue for future searches for new therapeutic approaches for treating or alleviating the symptoms of BD and SZ. Meanwhile, in infancy, when epilepsy usually begins, the apparent compensative mechanisms would weaken, with the pathogenic polymorphisms in *GAD1* ORF coming into focus and leading to dramatic decrease in the levels of GABA. We also hypothesize that the GABA vesicular transporter VGAT may be the most vulnerable element of the GABA pathway, as in contrast to GAD67-related epilepsy, predominantly associated with bi-allelic mutations; in case of VGAT, a heterozygous SNP is enough to cause the condition. Intriguingly, some of the pathogenic VGAT variants are characterized by excessive transport activity, thus enhancing GABA release and inhibition. Interestingly, no SZ-associated polymorphisms in VGAT encoding gene *SLC32A1* have yet been found, although VGAT level was decreased in both SZ and ASD. This decrease may result from both the existence of so far unidentified SNPs in *SLC32A1* regulatory elements or disruption of transcription factors regulating its expression. If no *SLC32A1* polymorphisms severe enough to contribute to SZ are ever found, this will most likely indicate that they are lethal. Regardless, additional research is needed to comprehend the role of VGAT in NDDs. Interestingly, a particular number of NDDs, mostly epileptic disorders, are associated with haploinsufficiencies of GABA transporter GATs—GAT1 regulating phasic inhibition, GAT3 regulating tonic inhibition, and BGT1 responsible for transporting betaine through the brain–blood barrier, thus indirectly modulating GAT1 function. The putative explanation for the clinical effect of single mutations in GAT1 implies misfolding and malfunctioning or subsequent degradation of half of the transporter’s molecules produced in the cell, the other half being insufficient for maintaining adequate levels of GABAergic inhibition. What needs to be clarified in this case is the mechanism of action of the three recently described SZ-associated GAT1 variants p.Ala93Thr, p.Arg211Cys, and p.Trp495Leu [203].

Speaking of the metabolic stage (except for the GABA synthesis step), it is reasonable to conclude that, in contrast to the signaling stage, its disruption has only a minor direct effect on GABA signaling and rather causes clinical effects due to accumulation of neurotoxic byproducts, such as excessive glutamate or GHB, with further mitochondrial depletion. Nevertheless, at least in the cerebellum, GLS1 is capable of indirectly enhancing the GABA pathway via the neuroinflammatory pathways TNFα–TNFR1–NF-κB [204]. Furthermore, neuroinflammation may be triggered by exposure to stress; for instance, Stattic, a drug known to downregulate IL-6 expression by inhibiting STAT3 phosphorylation at Tyr705, was shown to suppress epileptic activity in the hippocampal and limbic regions, accompanied by delayed and flat successive spike waves in a GEFS+ mouse model [205]. Moreover, in rats, chronic stress in adolescence, induced by the combination of restraint stress and footshock, and in adulthood caused SZ-like conditions including anxiety-like responses, disrupted cognitive function, and hyper-responsivity of DA neurons [206], the latter hinting at the possible role of stress in depleting the function of the GABA transporter GAT1 [151]. At the same time, acute stress (the threat-of-shock condition) resulted in approximately an 18% decrease in prefrontal GABA and presynaptic downregulation of GABAergic neurotransmission [207]. This indicates that acute and chronic stress affect the GABAergic neurotransmission via distinct mechanisms, but both may potentially accelerate the disease onset and/or aggravate its clinical symptoms in individuals with high genetic risk for psychiatric disorders. Hence, additional research on the interconnections between stress and the GABA pathway may be necessary for developing new therapeutic strategies for treating NDDs.

In sum, as research continues to refine the role of GABA pathway components in the pathogenesis of neurodevelopmental disorders, new therapeutic strategies may offer more effective and individualized interventions for patients with psychiatric disorders characterized by GABAergic dysfunction.

## Figures and Tables

**Figure 1 ijms-26-09492-f001:**
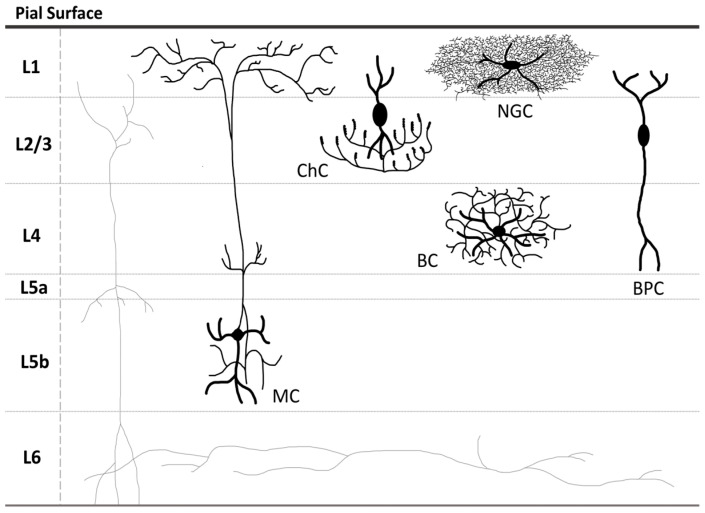
Localization of some types of inhibitory GABAergic neurons within the neocortex. MC—Martinotti cell; ChC—chandelier cell; BC—basket cell; NGC—neurogliaform cell; BPC—bipolar cell; L1, L2/3, L4, L5a, L5b, and L6—the neocortex layers.

**Figure 2 ijms-26-09492-f002:**
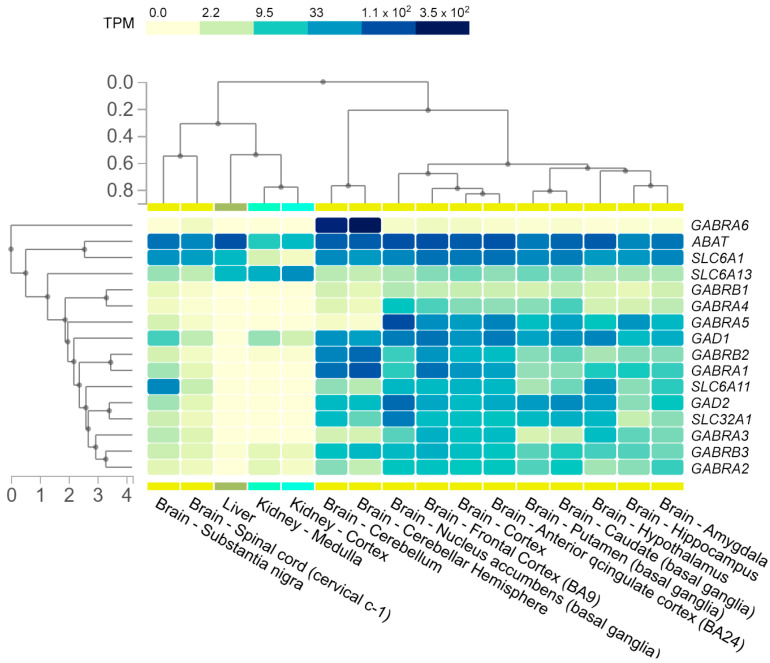
Expression of GABAergic system genes in various human brain regions and peripheral organs based on GTEx data (transcripts per million, TPM). The heatmap shows expression levels of key genes involved in the function of the GABAergic system, including genes encoding GABA_A_ and GABA_B_ receptor subunits (e.g., *GABRA1–6*, *GABRB1–3*, etc.), enzymes of GABA metabolism (*ABAT*, *GAD1*, *GAD2*), and GABA transporters (*SLC6A1*, *SLC6A11*, *SLC6A13*, *SLC32A1*). Tissues with negligible expression levels or not relevant to the focus of this study were not included in the analysis.

**Figure 3 ijms-26-09492-f003:**
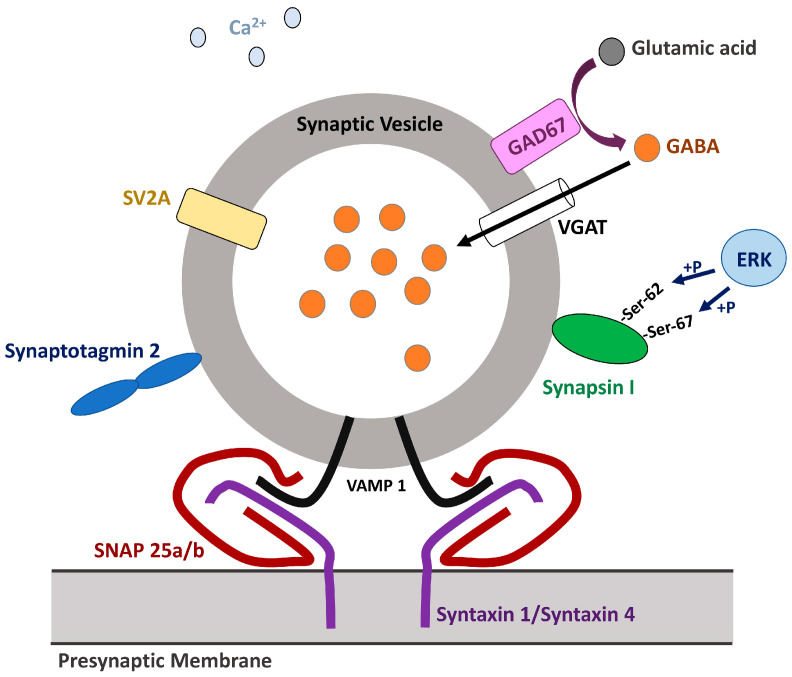
Interaction of the synaptic vesicle loaded with GABA with presynaptic membrane.

**Figure 4 ijms-26-09492-f004:**
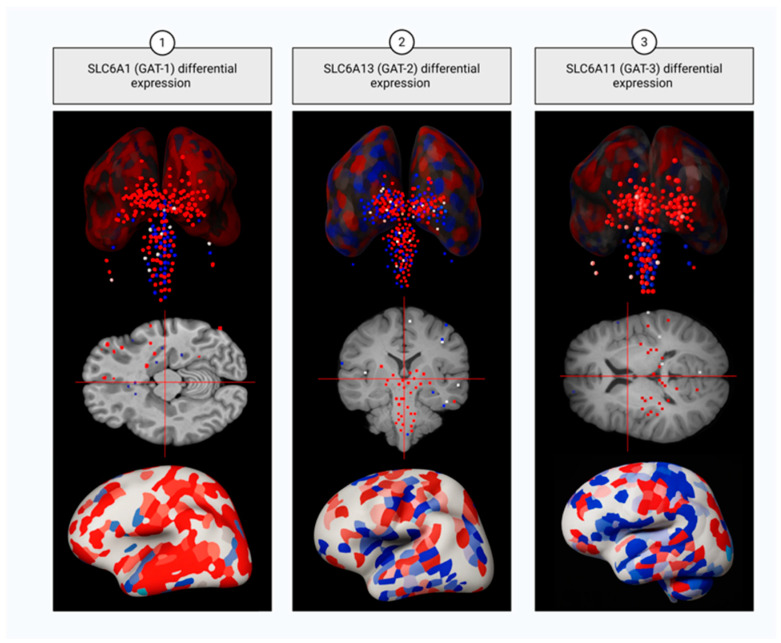
Expression of GABA transporter GAT1 (column 1), GAT2 (column 2), and GAT3 (column 3) in neocortex and subcortical structures. Top panels: regional expression values projected onto anatomical brain surfaces; middle panels: axial MRI slices illustrating spatial distribution; bottom panels: cortical surface renderings. Red denotes regions with upregulated expression, blue—regions with downregulated expression.

**Figure 5 ijms-26-09492-f005:**
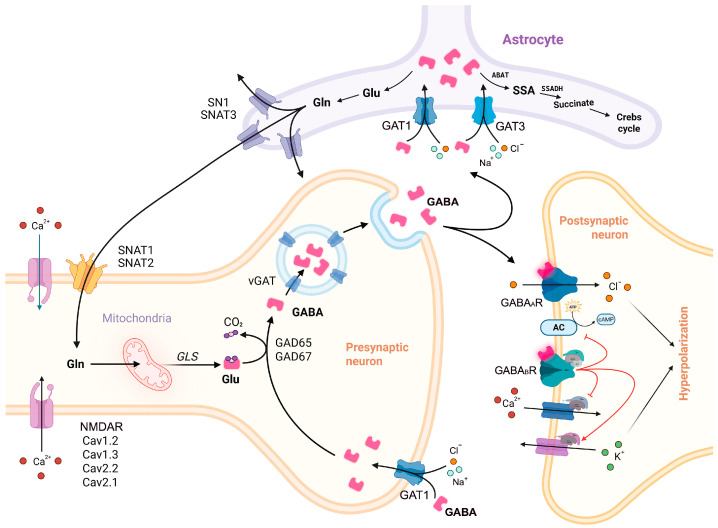
The GABA pathway. Glutamine (Gln) is released by the astrocyte via SNAT3 transporter and is further imported by SNAT1/2 into the GABAergic neuron, where it is metabolized to glutamic acid (Glu) by glutaminases GLS1/2. GAD67/65 converts Glu into GABA (pink polygons), which is then loaded into the SV by VGAT, trafficked to the presynaptic membrane, and released into the synaptic cleft. Once released, GABA interacts with its receptors GABA_A_R and GABA_B_R, activating Cl^–^ (orange circles) influx by the former, and modulating the activity of the potassium and calcium channels associated with the latter, increasing K^+^ (green circles) efflux and decreasing Ca^2+^ (red circles) influx; this, in turn, results in the postsynaptic membrane hyperpolarization. The excess of GABA is taken up by the astrocyte via GAT1 (or GAT3), which simultaneously translocates 2 Na^+^ ions (light-blue circles) and one Cl^–^ ion. In the astrocyte, GABA is sequentially catabolized by GABA transaminase ABAT and SSA dehydrogenase SSADH into succinic acid, which further enters the Krebs cycle, where it is processed into α-ketoglutarate, the precursor of Glu and Gln.

## Data Availability

The datasets generated during and/or analyzed during the current study are available from the corresponding author on reasonable request.

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
