# Peer review of "The Role of GABA Pathway Components in Pathogenesis of Neurodevelopmental Disorders"

_ijms, 2025, doi:10.3390/ijms26199492_

Round 1

Reviewer 1 Report

Comments and Suggestions for Authors

The review prepared by Marilovtseva and colleagues examines the literature on the role of several genes and proteins involved in the GABA pathway and the implication for the comprehension and treatment of neurodevelopmental disorders.

The manuscript is well written, raises important facts and proposes interesting take-home messages.  I am in favour of its publication

One important comment : why has Intellectual Disability (ID) not been considered ?

There is ample evidence of an involvement of the GABA pathway in ID, as well as evidence that Tiagabine could be used in the treatment of mild epilepsy and ID. 

Considering that ID is a feature common to many NDD (such as Down and Rett syndromes), this reviewer would ask that the authors mention, and perhaps include a paragraph on the role of GABA in ID.

A few published articles on the subject :

Epilepsy and intellectual disability linked protein Shrm4 interaction with GABABRs shapes inhibitory neurotransmission.

Zapata J, Moretto E, Hannan S, Murru L, Longatti A, Mazza D, Benedetti L, Fossati M, Heise C, Ponzoni L, Valnegri P, Braida D, Sala M, Francolini M, Hildebrand J, Kalscheuer V, Fanelli F, Sala C, Bettler B, Bassani S, Smart TG, Passafaro M. Nat Commun. 2017   8:14536. doi: 10.1038/ncomms14536. PMID: 28262662

Severe Intellectual Disability and Enhanced Gamma-Aminobutyric Acidergic Synaptogenesis in a Novel Model of Rare RASopathies.

Papale A, d'Isa R, Menna E, Cerovic M, Solari N, Hardingham N, Cambiaghi M, Cursi M, Barbacid M, Leocani L, Fasano S, Matteoli M, Brambilla R. Biol Psychiatry. 2017   81(3):179-192. doi: 10.1016/j.biopsych.2016.06.016.  PMID: 27587266

Tiagabine: a new therapeutic option for people with intellectual disability and partial epilepsy.

Kälviäinen R. J Intellect Disabil Res. 1998    42 Suppl 1:63-7. PMID: 10030435

3p25.3 microdeletion of GABA transporters SLC6A1 and SLC6A11 results in intellectual disability, epilepsy and stereotypic behavior.

Dikow N, Maas B, Karch S, Granzow M, Janssen JW, Jauch A, Hinderhofer K, Sutter C, Schubert-Bast S, Anderlid BM, Dallapiccola B, Van der Aa N, Moog U. Am J Med Genet A. 2014  164A(12):3061-8. doi: 10.1002/ajmg.a.36761. PMID: 25256099

GABA and glutamate pathways are spatially and developmentally affected in the brain of Mecp2-deficient mice.

El-Khoury R, Panayotis N, Matagne V, Ghata A, Villard L, Roux JC. PLoS One. 2014  9(3):e92169. doi: 10.1371/journal.pone.0092169. eCollection 2014. PMID: 24667344

GABAergic over-inhibition, a promising hypothesis for cognitive deficits in Down syndrome.

Zorrilla de San Martin J, Delabar JM, Bacci A, Potier MC. Free Radic Biol Med. 2018 Jan;114:33-39. doi: 10.1016/j.freeradbiomed.2017.10.002. PMID: 28993272.

Modulation of the GABAergic pathway for the treatment of fragile X syndrome.

Lozano R, Hare EB, Hagerman RJ. Neuropsychiatr Dis Treat. 2014 Sep 16;10:1769-79. doi: 10.2147/NDT.S42919. eCollection 2014. PMID: 25258535

Author Response

We thank the editor for thorough review of our paper and valuable comments.

Reviewer 1.

Comment: Why has Intellectual Disability (ID) not been considered?

Answer: We appreciate your recommendations and supplied the manuscript with additional details in accordance with them, using the papers you listed.

Interestingly, there is a number of other factors, including some signaling pathways, that govern GABA release. One of them is RAS–ERK signaling cascade, which is directly involved in pathogenesis of various health conditions called “RASopathies”, some of which manifest in ID and learning disability [88]. Recently, it has been shown that while in excitatory neurons hyperactivation of RAS–ERK signaling due to the SNPs (such as p.Gly12Val) within the KRAS is cytotoxic and leads to cell death, in GABAergic neurons it increases the risk of spontaneous GABA release [89]. The latter is, apparently, due to phosphorylation of synapsin I with its subsequent dissociation from SVs [90]. At physiological level, this results in increased sIPSC frequency, as well as impaired long-term potentiation at CA3–CA1 synapses in hippocampus that can be rescued by picrotoxin treatment [89]. So, it brings us to a conclusion that, when designing new diagnostic protocols and therapeutic agent for treating NDDs associated with dysregulated GABA traffic and release, the whole spectrum of its regulators should be taken into account.

Pathogenesis and clinical manifestation of a particular number of IDs, including those associated with chromosomal aberrations, are also known to be linked to impaired expression of GABAAR subunits. In Prader–Willi syndrome (PWS), a genetic disorder characterized by mild-to-moderate ID, impaired behavior, and depression [118], paternal alleles of GABRA5, GABRB3, and GABRG3 are missing due to the partial deletion of an imprinted region on paternal chromosome 15 (15q11–13) [119]. Given that all three GABAAR subunits are associated with hippocampus [99], the loss of a half of their alleles may explain poor short-term memory, another known symptom of PWS [120]. Furthermore, according to the in vivo data obtained by single–voxel proton magnetic resonance spectroscopy (1H-MRS), GABA levels are also reduced in the PWS brain (at least, in the parieto-occipital lobe) [121] hinting at the possibility of existence of a feedback loop regulating either GAD1 expression, or GAD67 enzymatic activity in GABAergic neurons in response to the fluctuations of GABAARs’ content in PyCs. Impaired GABA pathway is also typical for Fragile X syndrome (FXS), an ID caused by an expansion of the CGG repeat within the 5’-UTR of FRAGILE X MESSENGER RIBONUCLEOPROTEIN 1 coding gene FMR1 [122]. FXS mice models demonstrated 35–50% decrease in expression rates of Gabra1, Gabra3, Gabra4, Gabrb1, Gabrb2, Gabrg1, and Gabrg2 in the cortex [123], as well as a four-fold decrease in Gabrd subunit in the hippocampus [124]. Interestingly, [11C]flumazenil PET (position emission tomography) performed on FXS patients identified a significant reduction in the non-displaceable binding potential (BPND) of GABAARs (reflecting their availability), throughout the brain, with the highest level of BPND reduction – up to 17% – detected in the thalamus [125]. Dysregulation of several other components of the GABA pathway have also been found in association with FXS, GAD67/65, GABA transporters GAT1 and GAT4, and GABA catabolizing enzymes among them [122]. Noteworthy, in contrast to PWS and FXS, in Down syndrome (DS), GABA inhibition (mainly tonic) is enhanced. This phenomenon may be explained by the presence of an extra chromosome 21 and, hence, additional copies of OLIG1, OLIG2, and DYRK1A [126], well-known regulators of GABAergic interneurons production [127] and synaptic plasticity pathways [128]. Given the abovementioned data, it was suggested that treatment with basmisanil, a α5-GABAAR–specific negative allosteric modulator, could have a positive effect on cognition in patients with DS, but unfortunately, the clinical trials failed [129]. Nevertheless, therapeutic strategies targeting other components of the GABA pathway or the proteins, modulating GABA neurotransmission, may be effective in alleviating DS symptoms. One of the promising targets for such therapies is G-protein–activated inwardly rectifying potassium channel 2, GIRK2, whose gene is located on chromosome 21 and whose function is linked to that of GABAB receptors [130,131].

Finally, the actin-binding protein SHROOM4 (SHRM4) (encoded by KIAA1202), has recently been shown to play a role in regulating the receptor’s activity by providing its synaptic localization [140]. This finding makes GABABRs a key player in pathogenesis of Stocco Dos Santos syndrome [141] and the NDD caused by Xp11.2 microduplication [142], both of which are associated with disrupted KIAA1202 and characterized by mild or mild-to-severe ID, seizures, anxiety, and impaired behavior.

Finally, the recent studies have also demonstrated a moderate effect of 4-phenylbutyrate in reducing seizures (but not in mitigating cognitive impairment) in patients with SLC6A1 and SLC6A11 haploinsufficiency, caused by a microdeletion on 3p25.3 [164] and associated with ID, epilepsy, and multiple congenital abnormalities [165].

Reviewer 2 Report

Comments and Suggestions for Authors

Dear editor, I would like to thank you for inviting me to review the manuscript ijms-3883127, titled The role of GABA pathway components in pathogenesis of neurodevelopmental disorders'. Here the authors reviewed the crucial steps of GABA (γ-aminobutyric acid) synthesis, GABA receptors and how they transmit or secret GABA in synapses. Moreover, they described the process and the components involved in the reuptake of GABA and its eventual turn over. 

I commend the authors rigor in writing, the references they went through, and their suggestions in GABA metabolism's involvement in the neurodevelopmental disorders. However, I suggest that they should included a few more figures and/or tables which would benefit the readers. If they do so, I have no other concerns about this manuscript and will recommend its acceptance in your journal. 

There can be at least two figures in the sections where the authors are describing the classification GABA neurons and GABA synthesis and release. 

best, 

Author Response

We thank the editor for thorough review of our paper and valuable comments.

Reviewer 2

Comment: I suggest that they should included a few more figures and/or tables which would benefit the readers

Answer: Thank you for your revision. We appreciate your recommendation and added two extra figures to illustrate the first chapter of out manuscript (attached as a file)